# Antiviral Effect of 5′-Arylchalcogeno-3-aminothymidine Derivatives in SARS-CoV-2 Infection

**DOI:** 10.3390/molecules28186696

**Published:** 2023-09-19

**Authors:** Amanda Resende Tucci, Raquel Mello da Rosa, Alice Santos Rosa, Otávio Augusto Chaves, Vivian Neuza Santos Ferreira, Thamara Kelcya Fonseca Oliveira, Daniel Dias Coutinho Souza, Nathalia Roberto Resende Borba, Luciano Dornelles, Nayra Salazar Rocha, João Candido Pilar Mayer, João B. Teixeira da Rocha, Oscar Endrigo D. Rodrigues, Milene Dias Miranda

**Affiliations:** 1Laboratório de Morfologia e Morfogênese Viral, Instituto Oswaldo Cruz, Fundação Oswaldo Cruz, Rio de Janeiro 21041-250, RJ, Brazil; amanda.tucci@ioc.fiocruz.br (A.R.T.); alicerosa@aluno.fiocruz.br (A.S.R.); vivian.ferreira@ioc.fiocruz.br (V.N.S.F.); thamara.oliveira@ioc.fiocruz.br (T.K.F.O.); danielsouza@aluno.fiocruz.br (D.D.C.S.); 202003536733@alunos.estacio.br (N.R.R.B.); 2Programa de Pós-Graduação em Biologia Celular e Molecular, Instituto Oswaldo Cruz, Fundação Oswaldo Cruz, Rio de Janeiro 21041-250, RJ, Brazil; 3LabSelen-NanoBio—Departamento de Química, Universidade Federal de Santa Maria, Santa Maria 97105-900, RS, Brazil; raquel.mello@acad.ufsm.br (R.M.d.R.); luciano.dornelles@ufsm.br (L.D.); nayra.rocha@acad.ufsm.br (N.S.R.); joao.mayer@acad.ufsm.br (J.C.P.M.); 4CQC-IMS, Departamento de Química, Universidade de Coimbra, Rua Larga, 3004-535 Coimbra, Portugal; 5Laboratório de Imunofarmacologia, Centro de Pesquisa, Inovação e Vigilância em COVID-19 e Emergências Sanitárias (CPIV), Instituto Oswaldo Cruz (IOC), Fundação Oswaldo Cruz (Fiocruz), Rio de Janeiro 21040-900, RJ, Brazil; 6Programa de Pós-Graduação em Bioquímica Toxicológica, Universidade Federal de Santa Maria, Santa Maria 97105-900, RS, Brazil; joao.rocha@ufsm.br

**Keywords:** chalcogen-zidovudine, SARS-CoV-2, Calu-3 and Vero E6 Cell models, selenium, tellurium, viral replication, in silico calculations

## Abstract

The understanding that zidovudine (ZDV or azidothymidine, AZT) inhibits the RNA-dependent RNA polymerase (RdRp) of SARS-CoV-2 and that chalcogen atoms can increase the bioactivity and reduce the toxicity of AZT has directed our search for the discovery of novel potential anti-coronavirus compounds. Here, the antiviral activity of selenium and tellurium containing AZT derivatives in human type II pneumocytes cell model (Calu-3) and monkey kidney cells (Vero E6) infected with SARS-CoV-2, and their toxic effects on these cells, was evaluated. Cell viability analysis revealed that organoselenium (**R3a**–**R3e**) showed lower cytotoxicity than organotellurium (**R3f**, **R3n**–**R3q**), with CC_50_ ≥ 100 µM. The **R3b** and **R3e** were particularly noteworthy for inhibiting viral replication in both cell models and showed better selectivity index. In Vero E6, the EC_50_ values for **R3b** and **R3e** were 2.97 ± 0.62 µM and 1.99 ± 0.42 µM, respectively, while in Calu-3, concentrations of 3.82 ± 1.42 µM and 1.92 ± 0.43 µM (24 h treatment) and 1.33 ± 0.35 µM and 2.31 ± 0.54 µM (48 h) were observed, respectively. The molecular docking calculations were carried out to main protease (M^pro^), papain-like protease (PL^pro^), and RdRp following non-competitive, competitive, and allosteric inhibitory approaches. The in silico results suggested that the organoselenium is a potential non-competitive inhibitor of RdRp, interacting in the allosteric cavity located in the palm region. Overall, the cell-based results indicated that the chalcogen-zidovudine derivatives were more potent than AZT in inhibiting SARS-CoV-2 replication and that the compounds **R3b** and **R3e** play an important inhibitory role, expanding the knowledge about the promising therapeutic capacity of organoselenium against COVID-19.

## 1. Introduction

COVID-19, caused by the highly pathogenic novel coronavirus (CoV) known as SARS-CoV-2, was characterized as a pandemic by the World Health Organization in March 2020 [1]. Since the outbreak began in December 2019, SARS-CoV-2 has spread from China to the rest of the world. Currently, it is estimated that over 770 million cases of infection have been reported since the beginning of the pandemic, and over 6.9 million deaths have been reported worldwide [2]. Globally, disease cases continue to rise, and therefore the development of effective antiviral therapies to aid severe SARS-CoV-2 infected patients is relevant [3,4].

The clinical scenario of COVID-19 is variable, and the classic symptoms are compatible with those of influenza syndromes during the initial stages of infection [5]. However, about 10–15% of symptoms progress to acute pneumonia, pulmonary embolism, cardiomyopathy, and disseminated intravascular coagulation, which pose a serious risk of mortality in these cases [6,7,8,9]. It is recognized that a progressive infection can lead to exacerbated immune and inflammatory responses [10]. Therefore, different therapeutic strategies using antivirals, antiretrovirals, antimalarials, and anti-inflammatory drugs, as well as corticosteroids, immunomodulators, or immunoglobulin therapies, have been required and repurposed for the COVID-19 treatment [11,12]. It is important to highlight that the effectiveness of vaccines has been fundamental in preventing new cases and improving the symptomatology of the disease. However, there are still significant challenges when it comes to developing specific antivirals against SARS-CoV-2 [13,14,15]. 

SARS-CoV-2 is a β-coronavirus belonging to the family Coronaviridae and containing a single-stranded positive polarity RNA (+ssRNA) [13,14]. The genome encodes four structural proteins, which play an important role in the assembly of virions and in the activation of the host’s immune response, and are classified as nucleocapsid (*N*), spike (*S*), membrane (*M*), and envelope (*E*) proteins, and sixteen non-structural proteins (nsp1-16) that are essential for viral transcription and replication [16,17]. The main protease (M^pro^ or 3CLpro), papain-like protease (PL^pro^), and RNA-dependent RNA polymerase (RdRp) are configured as non-structural proteins and have been the focus of numerous research studies and stand out as key molecules in the fight against COVID-19 [18,19,20,21,22].

The approved drugs and drugs authorized under an emergency use by the U.S. Food and Drug Administration (FDA) are characterized as protease inhibitors (nirmatrelvir/ritonavir), RdRp inhibitors (remdesivir and molnupiravir), inflammatory mediators (baracitinib and tocilizumab), and monoclonal antibodies against SARS-CoV-2, representing a significant advancement in the fight against this virus [23]. However, clinical studies have shown limited or nonexistent efficacy for most proposed medications, as well as it was reported various side effects [24,25]. Additionally, there is the occurrence of COVID-19 rebound effect with the treatment based on molnupiravir and nirmatrelvir/ritonavir [26]. Furthermore, the emergence of new strains that are more transmissible and have higher infectivity is referenced as another concerning factor, and thus the development of new drugs is an important strategy to broaden therapeutic targets and consequently reduce the emergence of new mutations in the virus [15]. In this context, some organochalcogenium compounds have been described in the literature as showing anti-SARS-CoV-2 activity. For instance, Mangiavacchi et al. reported the seleno-functionalization of quercetin and the corresponding anti-SARS-CoV-2 activity. The authors described that the introduction of the specific *para*-tolylselenyl fragment at the quercetin moiety led to a 24-fold improvement in the antiviral potency when compared with the original quercetin [27]. Additionally, at the beginning of the SARS-CoV-2 spread, it was reported the potential application of ebselen as an anti-SARS-CoV-2 agent to target the viral proteases [28,29], and recently different groups have reported novel ebselen derivatives to improve the anti-SARS-CoV-2 profile, at interacting not only with M^pro^ or PL^pro^ but also with nsp14 guanine N7-methyltransferase and RdRp [30,31,32,33].

An important trend in the chemotherapy field involves the study of nucleoside analogs and the analysis of their multiple antioxidant, antitumor, antimicrobial, and antiviral properties [34,35,36,37]. Azidothymidine (AZT), also known as zidovudine (ZDV), is a synthetic nucleoside analog of thymidine that was originally synthesized as an antitumor compound but gained notable prominence due to its inhibitory activity on reverse transcriptase, an essential enzyme for HIV replication [38,39]. Interestingly, current studies have revealed that AZT has also been described as an inhibitor of SARS-CoV-2 RNA-dependent RNA polymerase (RdRp) in vitro (kinetic test) [40,41]. In silico analyses have shown that the antiretroviral drug binds to the catalytic site of RdRp and impairs the functioning of the transcriptase-replicase complex responsible for the formation of new viral RNA strands [42]. However, since prolonged use of AZT has been correlated with the emergence of severe side effects due to its high toxicity [38,43], the development of analog molecules, based on AZT scaffold, has been required as a strategy for new therapies against cancer and viral infections [44,45].

Previous data from the group have demonstrated that new compounds formed by adding chalcogen atoms to the structure of commercial AZT were able to enhance the in vitro antioxidant and antitumor action of the hydride molecules [46,47]. In this context, it was reported that the presence of chalcogen in the molecule 5′-arylchalcogenyl-3-(phenylselanyl-triazoyl)-thymidine reduced the proliferative capacity of bladder cancer cells and amplified the activity of the compound [48]. Additional results revealed that in breast cancer cell line, AZT analogs containing tellurium were more effective in inhibiting tumor growth and showed lower cytotoxicity than the original antiretroviral molecule [46]. Furthermore, it was described that the organochalcogen 5′-selenothymidine (S1073) acts as an important neuroprotective agent and attenuates oxidative stress in mice with cognitive dysfunction induced by intracerebroventricular-streptozotocin (ICV-STZ) [49]. Expanding on these findings, Ecker et al. [50,51] indicated that, after selenium insertion, the compound 5′-(4-methylphenylseleno) zidovudine (SZ3) was not toxic to human peripheral blood mononuclear cells (PBMCs) and exhibited a protective action on red blood cells. In summary, these data expand our knowledge of nucleoside analogs, reveal the multifunctional role of AZT-derivatives, and direct us towards the search for new low-toxicity chalcogen-zidovudines with potent antiviral activity against SARS-CoV-2.

Thus, in the present study, human pneumocytes type II model (Calu-3) and monkey kidney cells (Vero E6) were infected with novel coronavirus, and the inhibitory capacity of organochalcogens from the chalcogen-zidovudines (5′-arylchalcogeno-3-aminothymidine derivatives, Figure 1) was evaluated against in vitro viral replication. In addition, molecular docking calculations were carried out to evaluate the capacity of these chalcogen-zidovudine derivatives to interact with the three main enzymes reported as key targets of this class of compounds (M^pro^, PL^pro^, and RdRp). Since the experimental enzymatic assays were not conducted, it was considered non-competitive, competitive, and allosteric inhibitory mechanisms for the in silico approach for a better interpretation of the interaction between drug and target.

## 2. Results and Discussion

### 2.1. Synthesis of 5′-Arylchalcogeno-3-aminothymidine Derivatives

The compounds were prepared by LabSelen NanoBio Laboratory in the Chemistry Department at Federal University of Santa Maria—RS, as described by Da Rosa [52]. Briefly, AZT was initially mesylated employing mesyl chloride and Et_3_N, producing the respective AZT-mesylate **1** (Figure 1). Then, the chalcogenium portion was introduced via chalcogenolate anion, obtained by reaction of diaryldichalcogenide **2** and NaBH_4_. The azide reduction was performed in the same reduction system (Figure 1). 

A total of 12 compounds among selenides (**R3a**–**R3e**), tellurides (**R3f**, **R3n**–**R3q**), sulfide (**R3r**), and AZT were obtained (Figure 1) with yield in the range of 40–78% (see Appendix A). The respective compounds were characterized using nuclear magnetic resonance (NMR) techniques (^1^H and ^13^C) and high-resolution mass spectrometry (HRMS) analysis. Additional information on experimental procedures, NMR data, and HRMS analysis are in da Rosa et al. and Leal et al. [52,53].

### 2.2. The Effect of Chalcogeno-Zidovudines on Different Cell Lineages Viability

The 5′-arylchalcogeno-3-aminothymidine derivatives (**R3** series), previously recognized by the group as important antioxidant and antitumor agents with broad biological spectrum properties [52], were evaluated in this study as potential inhibitors of SARS-CoV-2 replication. The antiviral activity of the compounds was evaluated in two distinct cell lines: Calu-3 (that recapitulate human type II pneumocytes) and Vero E6 (African green monkey kidney cells). These cellular models are well-established in studies of SARS-CoV-2 infection. However, the mechanisms of interaction of the Spike protein with the proteases of each of these host cells and the intercellular route of coronavirus are distinct [54,55]. The Calu-3 represents the better infection model, since SARS-CoV-2 enters in host cell mediated by transmembrane protease serine 2 (TMPRSS2) [56]. This human pneumocyte cell line is widely used as a preclinical model of respiratory cells for drug screening and nasal spray development against respiratory infections, thus was used in this research as the main in vitro infection model [57,58,59].

First, we assayed the toxicity of the organocompounds to ensure the safety before the antiviral assays. Therefore, non-infected cells were treated with high concentrations (6.25–200 µM) of each chalcogen-zidovudine derivative (**R3** series) and subsequently processed for the cell viability assay.

The cell toxicity was determined by methylene blue procedure and the results showed that all chalcogen-zidovudine derivatives containing selenium (**R3a**–**3e**) stood out in the cell-based analysis for presenting CC_50_ values ≥ 100 µM for both Calu-3 and Vero E6 models (Figure 2A,C). Exclusively, the **R3a**–**3c** showed CC_50_ > 200 µM in Calu-3 and Vero E6 cells (Table 1). In contrast, it was evidenced that despite tellurium compounds (**R3f**; **R3n**–**R3q**) being more cell toxic than the selenium (**R3a**–**3e**) compounds, the tellurium compounds are not toxic to either cell models at the maximum concentration used (10 µM) in the anti-SARS-CoV-2 assays (Section 2.3). The viability of Vero E6 and Calu-3 was substantially reduced after the treatment with the compounds **R3f** and **R3n**–**R3q** at high concentrations (Figure 2B,D). On average, CC_50_ values for organoselenium compounds are up to 6-fold greater than those for organotellurium compounds (Table 1). The organosulfur (**R3r**) and AZT were used as compound controls, and the CC_50_ values were >200 µM in Calu-3 for both compounds.

The understanding that nucleoside and nucleotide analogs are important therapeutic agents and are widely used in the treatment of viral diseases and cancer [60] has directed our research towards molecules belonging to this class and their possible role in the fight against COVID-19. AZT, originally synthesized as an antitumor compound [61], has been approved by the FDA and categorized as an important antiretroviral in the treatment of AIDS [62]. Although AZT plays a valuable role as an inhibitor of the reverse transcriptase of the HIV virus, it also exhibits antibacterial activity in vivo [63] and inhibits the growth of breast, ovarian, and lung tumors [64,65,66]. However, there are clinical reports that AZT causes many side effects and prolonged use can lead to bone marrow toxicity, causing anemia and leukopenia, and even psychosis, myopathy, cardiopathy, and hepatotoxicity [43,67,68]. Therefore, modifications in the structure of AZT aiming to reduce the drug’s toxicity without compromising its biological activity have been proposed [52,69], e.g., data previously generated by the group reported that the addition of chalcogen in the AZT core was able to amplify the antitumor and antioxidant activity of the new compounds, as well as reducing cell damage and toxicity in treatments performed with human leukocytes and in mice inoculated with the organochalcogen [52].

Synthetic compounds containing chalcogen atoms have been the subject of several studies due to their broad biological properties [70,71]. The performance of organoselenium, organotellurium, and organosulfur against free radicals, as well as their effectiveness as antiviral, anti-inflammatory, and antitumor agents, has directed many scientific studies [72,73,74]. Rocha et al. [46], in their research on breast cancer, considered that AZT combined with tellurium not only exhibited selectivity between cancerous and healthy cells, but also had a good pharmacokinetic profile and efficient protective action against oxidative stress. Furthermore, in the study with derivatives of 3′-triazolyl-5′-arylchalcogenilthymidine containing tellurium, it was observed that the higher reactivity and electron-donating capacity presented by this chalcogen justified its better anti-proliferative performance under bladder carcinoma 5637 cells, and, contrary to expectations, tellurium compounds were less toxic in human cell lines and rodents than selenium compounds [75].

The toxicity presented by selenium and tellurium compounds has been debated by some authors, and although studies have shown that organotellurium compounds are less toxic than selenium derivatives [74,75], consistent data indicate that organoselenium compounds have reduced toxicity [50,76,77,78,79]. In the present analyses, it was observed that Vero E6 and Calu-3 were preserved in a selenium organocompounds exposition, and that AZT derivatives containing tellurium were highly toxic for the cells at higher concentrations treatment. In addition to this analysis, it is interesting to note that other selenium compounds also presented low cytotoxicity in other human cell lineages, such as 5′-(4-methylphenylseleno)zidovudine (SZ3), which showed reduced toxic effect on immune human cells [51]. Therefore, these findings suggest that the chalcogen-zidovudine activity may be broad, and that this compound class can be categorized as an interesting candidate for future studies as an antiviral.

### 2.3. Anti-SARS-CoV-2 Activity of Chalcogeno-Zidovudines

The recognition that chalcogen addition to the AZT structure increases its bioactivities [80] and that AZT inhibits the RdRp of SARS-CoV-2 [40,41,42] led us to analyze the possible antiviral effect of 5′-arylchalcogeno-3-aminothymidine derivatives on the SARS-CoV-2 replication in previously infected Vero E6 and Calu-3 cells. The concentration used in cytotoxicity assays was up to 20-fold higher than the concentration used in antiviral assays, which ensures the cell viability in this study.

Here, we evaluated the activity of 1**2** nucleoside analogs belonging to the class of organoselenium (**R3a**, **R3b**, **R3c**, **R3d** and **R3e**), organotellurium (**R3f**, **R3n**, **R3o**, **R3p** and **R3q**), organosulfur (**R3r**), and AZT (Figure 1), focusing on molecules that showed the best performance against SARS-CoV-2 in Calu-3 cells. Therefore, the most promising molecules were those that presented high effectivity with lower EC_50_ and higher selectivity index (SI; calculated by the ratio of CC_50_ and EC_50_ values).

The obtained results demonstrated that at 24 h post-infection (hpi) in Vero E6 cells, compounds **R3e**, **R3f**, and **R3p** inhibited approximately 95% of SARS-CoV-2 replication at the highest tested concentration (10 µM) (Figure 3). However, the most interesting EC_50_ values were notable for organoselenium molecules (**R3a**–**R3e**) and **R3f**, reaching values below 3.0 µM (Table 1). The pharmacological activity of these compounds against SARS-CoV-2 replication is comparable to that presented by antiretroviral atazanavir in infected Vero E6 cells (EC_50_: 2.0 ± 0.12 µM) [81].

In human type II pneumocytes cells (Calu-3), widely used as a model for human respiratory diseases study and new drugs discovery [58], the treatment effect was evaluated in 24 and 48 hpi. The results obtained after 24 hpi showed that the **R3e**, **R3f**, **R3n**, and **R3q** at 10 µM inhibited coronavirus replication by over 95% with EC_50_ ≤ 2 µM (Figure 4A,B and Table 1). On the other hand, when the treatment period was extended to 48 hpi, only the **R3b** was able to inhibit viral replication by 95% at 10 µM with EC_50_ value of 1.33 ± 0.35 µM (Figure 4C). This result is comparable those presented for daclatasvir, a SARS-CoV-2 RdRp inhibitor (EC_50_: 1.1 ± 0.3 µM), on Calu-3-based assays also after 48 hpi [82]. Furthermore, selectivity index (SI) data indicated significant activity of this molecule at a later treatment period, without harming the viability of the host cell. In addition, it was observed that the EC_50_ for **R3d**, **R3e**, **R3n**, **R3o**, and **R3q** was ≤2.57 µM (Table 1). However, the EC_50_ for **R3d** and **R3o** did not represent the concentration of molecules necessary to inhibit 50% of viral replication, because 100% of virus replication inhibition at the maximum evaluated concentration (10 µM) was not observed. Overall, the data highlighted that the selectivity indexes for organoselenium molecules were up to 6-fold higher than for organotellurium compounds (Table 1). Organosulfur (**R3r**) and AZT were used as compound controls for structural comparison, showing low antiviral effect in the analyzed cell model if compared to the organoselenium compounds tested (Figure 4B,C, Table 1). 

To validate the experimental assays, atazanavir was used as experimental control for the cell toxicity and the virus inhibition at its CC_50_ and EC_50_ values in Vero E6 and Calu-3 assays [81,83].

Interestingly, only organoselenium **R3e** presented low and similar EC_50_ values in all cell types tested and regardless of the established treatment time, as verified in Calu-3 cells (Table 1). Moreover, their effect was equivalent to molnupiravir (estimated EC_50_ of 1.97 µM), an FDA-approved drug for COVID-19 emergency use [84], and more potent than lopinavir/ritonavir (EC_50_: 8.2 ± 0.3 µM), proposed as a treatment for COVID-19 during 2020 [82]. Furthermore, the selenium derivatives showed prominent SI values, presenting high CC_50_ values in antiviral in vitro assays. These results contrasted with the cell viability observed in cultures treated with AZT derivatives containing tellurium. In this context, it is important to emphasize that although tellurium compounds have shown interesting EC_50_ values in both infection models, the curves’ R^2^ were lower than for organoselenium molecules. Organotellurane’s stability in aqueous solution, such as culture medium, has already been reported [85]. However, we observed a notable results variation. This fact, associated with the high toxicity exhibited by these molecules and their reduced selectivity index (Table 1), directly impacts the non-choice of chalcogen-zidovudines containing tellurium as promising candidates for the next in silico assays.

Repurposing drugs is an important strategy for the development of new effective therapies against SARS-CoV-2 [86]. Remdesivir, an FDA approved drug for COVID-19 treatment, has paved the way for other repurposing drugs [23]. The remdesivir mechanism of action was originally described against positive-strand RNA viruses, including Ebola, HCV, MERS-CoV, and SARS-CoV [87,88,89], with inhibition of viral RdRp [90,91]. Additional studies have shown that sofosbuvir, alovudine, and zidovudine also act as SARS-CoV-2 polymerase inhibitors [40,92]. In addition, the pharmacological importance of organic selenium compounds against SARS-CoV-2, including the ebselen (Eb) and diphenyl diselenide ((PhSe)_2_), has been highlighted since the beginning of the COVID-19 pandemic [33,93,94,95]. The EC_50_ values obtained for tested organoselenium molecules are comparable to those already reported for Eb and ((PhSe)_2_ in Calu-3, for both 24 and 48 hpi [94,96]. The obtained data become even more promising due to the solid understanding that selenium derivatives might interact with key enzymes for SARS-CoV-2 replication, i.e., M^pro^, PL^pro^, and RdRp [29,30,31,32,33]. In addition, selenium derivatives also play a key role in the host immune response against RNA viruses, as well as contributing to maintaining redox homeostasis and modulating the inflammatory response, being a feasible multi-target compound [97]. Thus, motivated by the possible interactive profile between selenium derivatives and SARS-CoV-2 M^pro^, PL^pro^, and RdRp, molecular docking calculations were carried out to suggest the main target that the chalcogen-zidovudine derivatives containing selenium compounds (**R3a**–**R3e**) might interact with. **R3r** (chalcogen-zidovudine with sulfur) and AZT were used as comparative molecules. Finally, the possible inhibitory mechanism (competitive, non-competitive, or allosteric) was also explored via in silico calculations.

### 2.4. In Silico Studies

The favorable in vitro results as anti-SARS-CoV-2 for the chalcogen-zidovudine derivatives containing selenium compounds (**R3a**–**R3e**) led to the in silico evaluation on the main targets that these compounds might interact (M^pro^, PL^pro^, and RdRp). Additionally, it was also evaluated the compound **R3r** to verify the selenium effect on the binding capacity, while AZT (in the active form, AZT triphosphate—AZTTP) was used as a positive control for RdRp. Table 2 summarizes the docking score value (dimensionless) for each enzyme in different inhibitory mechanisms approaches. Since each pose obtained by GOLD 2022.3 software is considered as the negative value of the sum of energy terms, a more positive docking score value indicates better interaction [98].

For both SARS-CoV-2 proteases, **R3a**–**R3e** and **R3r** did not show favorable docking score values into the catalytic site in the presence of substrate, suggesting a competitive or allosteric inhibitory approach. The 3D structure of M^pro^ is composed of a homodimer and its active site is characterized by the presence of amino acid residues His-41 and Cys-145 (Figure 5A), while PL^pro^ is a homotrimer with Cys-112, His-273, and Asp-287 residues as a catalytic triad (Figure 5B). In addition, for M^pro^ there are two allosteric sites: one in the interface between both monomers formed by Glu-288, Asp-289, Phe-290, and Thr-291 residues (ALST1), and the other on each chain A and B formed by Lys-12, Cys-16, and Lys-97 (ALST2) (Figure 5A) [99,100,101]. The docking score values for both SARS-CoV-2 proteases by competitive inhibitory mechanism approach are quite similar, suggesting that chalcogen-zidovudine derivatives containing selenium compounds can interact with both M^pro^ and PL^pro^. These results agree with previous literature on the inhibitory capacity of selenium compounds to SARS-CoV-2 proteases [30,31,102,103].

Since the evaluated compounds have AZT moiety, the SARS-CoV-2 RdRp can also be considered as a feasible target to **R3a**–**R3e**. Interestingly, the docking score values for RdRp into the catalytic site without and in the presence of substrate are lower than for the allosteric site located in the palm region (ALST1) (Table 2 and Figure 5C). Additionally, the docking score value for the palm region is higher than for SARS-CoV-2 proteases, suggesting that the chalcogen-zidovudine derivatives containing selenium compounds interact preferentially with RdRp than M^pro^ and PL^pro^ via ALST1. The docking score value for AZT-TP (positive control) is higher than for **R3a**–**R3e** into the catalytic site of RdRp in the presence of substrate. It occurs due to the phosphorylation of AZT structure by endogenous enzymes (prodrug), acting as a chain terminator [99,100,104]. Despite **R3a**–**R3e** having AZT moiety, it is difficult to ascertain if they act similarly to AZT-TP mainly due to the low probability of chalcogen-zidovudine derivatives containing selenium compounds to be phosphorylated by endogenous enzymes. A future combination of biochemical and biophysical assays, e.g., experimental enzymatic inhibitory assays, thermal shift, and surface plasmon resonance, should be performed to further clarify how **R3a**–**R3e** could target SARS-CoV-2 RdRp.

Molecular docking results identified different binding poses and connecting points between each chalcogen-zidovudine derivative containing selenium and RdRp into ALST1 (Figure 5D,E). Hydrophobic and hydrogen bonding were detected as the main intermolecular forces (Figure 5F–K and Table 3); however, the *p*-methylbenzene and naphthalene moieties from **R3b** and **R3e**, respectively, might interact via π-stacking forces with His-439 residue in a positive electrostatic pocket of the enzyme (Figure 5E,G,J). This has been reported as an important region for the SARS-CoV-2 RdRp activity [105]; thus, this might be one of the reasons that the compounds **R3b** and **R3e** had better in vitro anti-SARS-CoV-2 results than the others tested compounds.

It is important to highlight that despite the replacement of selenium (**R3a**) by sulfur (**R3r**) did not impact drastically the docking score value, which was expected mainly due to their structural similarities, the sulfur-bearing **R3r** also did not display considerable inhibition of viral replication in vitro, probably due to the impact on the binding pose, resulting in differences in the interactive profile with the amino acid residues located in the ALST1 of SARS-CoV-2 RdRp. Additionally, besides the possible influence of the size of chalcogen atoms, the selenium- and tellurium-related redox processes may play a key role in the viral inhibition activity [33,106], reinforcing their differences in the experimental assays. This is supported by extant literature, as selenium- and tellurium-bearing organic compounds may undergo reversible redox reactions in biological medium [79,107], leading to either anti- or prooxidant effects upon enzymes, particularly thiol-dependent ones [108,109]. As selenoethers and telluroethers are among the known biologically active organochalcogen derivatives [110,111], the redox-related hypothesis may deserve proper evaluation in future studies.

## 3. Materials and Methods

### 3.1. Cell Culture and Virus

African green monkey kidney cells (Vero E6, ATCC CRL-1586) and human type II pneumocytes model (Calu-3) were kindly provided by Farmanguinhos platform RPT11M and grown in high-glucose Dulbecco’s modified Eagle medium (DMEM; GIBCO) supplemented with 10% fetal bovine serum (FBS), 100 U/mL penicillin and 100 µg/mL streptomycin. The cells were grown in 96-well plates with a density of 1.5 × 10^4^ cells/well in an incubator at 37 °C with 5% CO_2_ atmosphere. Afterwards, the cultures were processed for different assays. The SARS-CoV-2 isolate (lineage B.1, GenBank #MT710714, SisGen AC58AE2) was stored at −80 °C and manipulated in a biosafety level 3 (BSL3) environment, in accordance with World Health Organization recommendations [112].

### 3.2. Cytotoxicity Assays

Vero E6 and Calu-3 cells grown in 96-well plates were treated with molecules for 72 h at different concentrations, ranging from 6.25–200 µM. The compounds were previously solubilized in 100% dimethylsulfoxide (DMSO) and reached a final concentration of 0.1% (*v*/*v*) after dilution in DMEM in order to not compromise cell growth [113,114]. As an experimental control, the cell lines were maintained only with the solvent DMSO in the same proportion of treated cells.

The cell viability was determined by staining with methylene blue to quantify the percentage of viable cells post-treatment [115]. For this analysis, the cells were washed with saline solution and fixed/stained for 1 h in the incubator with methylene blue solution (Hanks’ solution (HBSS) + 1.25% glutaraldehyde + 0.6% methylene blue). Then, the solution was removed, and cells were washed with distilled water and dried at room temperature (RT). Subsequently, elution solution (50% ethanol, 49% PBS, 1% acetic acid) was added and the cells were incubated for 15 min at RT. Finally, the supernatant of the stained cultures was transferred to a 96-well plate and read at a wavelength of 660 nm. To evaluate the data obtained, a viability curve was constructed to obtain the CC_50_ values, which is the molecule concentration that causes the death of 50% of the treated cells [115].

### 3.3. Viral Replication Inhibition Assays

Vero E6 and Calu-3 cells were infected with SARS-CoV-2 at a multiplicity of infection (MOI) of 0.01 for 1 h at 37 °C. The cells were treated with the **R3** series molecules using a concentration curve (0.6, 1.25, 2.5, 5 and 10 µM). After 24 and 48 hpi, the supernatants were harvested, and the virus was grown in the presence or absence of molecules titrated. The effect of the treatment was evaluated by comparing the cells that were only infected, which were characterized as the infection control, with infected and treated cells. The EC_50_ values obtained are the concentration of molecule necessary to obtain 50% of its effective inhibitory activity [96,115]. The CC_50_ and EC_50_ values of the atazanavir (ATV) were used as an experimental control for virus inhibition and cell viability in Vero E6 and Calu-3 assays [81,83].

### 3.4. SARS-CoV-2 Titration

The replication capacity of SARS-CoV-2 in cell cultures with or without treatment was performed by plaque forming units’ assays (PFU/mL). Vero E6 previously seeded in 96-well plates were exposed to different dilutions of the supernatants. After 1 h of infection, 2.4% carboxymethylcellulose medium (containing DMEM 10×, sodium bicarbonate 0.22%, FBS 2%, penicillin 1%, and streptomycin 1%) was added and the infection was maintained for 72 h at 37 °C with 5% CO_2_ atmosphere. For the control, non-exposed cells to virus supernatants followed the same steps described above. After the infection time, the same volume of formalin 10% was added for cell fixation and viral inactivation. After 3 h, the medium was removed, and the monolayer was stained for 1 h with crystal violet 0.04%. After this step, the dye was removed, the wells were washed in running water, and dried at RT for subsequent quantification of PFUs [96,115].

### 3.5. Graphics

The graphics were made using GraphPad Prism version 9.0 program for Windows (GraphPad Software, Boston, MA, USA). The CC_50_ and EC_50_ values were determined by Nonlinear regression of Log (inhibitor) vs. Normalized response, of best curve generated. 

### 3.6. Molecular Docking Studies

Structure models of compounds **R3a**–**R3f**, **R3r**, and AZT (in the active form, AZT triphosphate—AZTTP) were built in Avogadro molecular editor [15], followed by energy minimization by UFF method, available in the same software [116]. In all examples, amine was in a protonated state to conform with biological medium. Energy-minimized models were further optimized by PM6 semiempirical method [117], included in MOPAC quantum chemistry software, version 2016. The crystallographic structure of M^pro^, PL^pro^, and RdRp was obtained in the Protein Data Bank (PDB), with access code 6LU7, 6W9C, and 7BV2, respectively [118,119]. Molecular docking calculations were performed with GOLD 2022.3.0 software (CCDC, Cambridge Crystallographic Data Centre, Cambridge, UK) [120] considering competitive, non-competitive, and allosteric inhibitory mechanisms. A radius of 10 Å around the corresponding binding sites was delimitated for molecular docking calculations and the standard scoring function “ChemPLP” was used. The commercial substrate of M^pro^ (S_Mpro_, modified peptide DABCYLKTSAVLQSGFRKME-EDANS with CAS number 730985-86-1) and PL^pro^ (S_PLpro_, modified peptide Z-RLRGG-AMC with CAS number 167698-69-3) was docked in the active site of chain A of the corresponding protease, and the best result was replicated for the other chains following previous publication [121]. On the other hand, the substrate of RdRp was reported in the crystallographic structure of this enzyme. Protein-Ligand Interaction Profiler (PLIP) webserver (https://plip-tool.biotec.tu-dresden.de/plip-web/plip/index, accessed on 23 August 2023) [122] was used for the identification of protein-ligand interactions and the 3D-figures were generated by PyMOL Molecular Graphics System 1.0 level software (Delano Scientific LLC software, Schrödinger, New York, NY, USA) [123].

## 4. Conclusions

Cell-based data have shown that AZT derivatives containing selenium outperformed tellurium compounds in anti-SARS-CoV-2 screening. The selenium molecules **R3b** and **R3e** stood out for their potent effect on the SARS-CoV-2 replication in Calu-3 and Vero E6 cells, with low cytotoxicity. Furthermore, the results have expanded the possibilities of in silico studies with the main molecular targets for coronavirus replication (M^pro^, PL^pro^, and RdRp). Although more efforts are needed to understand the molecular mechanisms of **R3b** and **R3e** action, the in silico results suggested that these molecules mainly interact in the allosteric site of RdRp located in the palm region. Moreover, the role of selenium as an inflammation modulator also opens ways for further research on the benefits of chalcogen-zidovudine derivatives in the COVID-19 pathogenesis.

## Data Availability

Not applicable.

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
