# Peer review of "Antiviral Effect of 5′-Arylchalcogeno-3-aminothymidine Derivatives in SARS-CoV-2 Infection"

_molecules, 2023, doi:10.3390/molecules28186696_

Round 1

Reviewer 1 Report

1.      In the abstract, selenium and tellurium containing AZT analogues,...whereas the title describe the derivatives, the title and abstract and rest of sentences throughout the manuscript must be aligned

2.      zidovudine (AZT), AZT abbreviation is not well suited, change it

3.      Abstract. The introductory part was complete ignored.

4.      The methods shall be clearly explain, the results as well, the results shall be quantitative. Then draw the conclusion in connection with the results

5.      Seleno-Functionalization of Quercetin, do not use uppercase throughout the manuscript

6.       The rational of the study is very weak, the literature reported analogs of same type were not incorporated for the comparison. What is novelty of this study, no explanations, and the advantages of present study over the reported study need to explain?

7.      Discussion is weaker and conclusion was not draw to supported the present study

8.      Experimental, nothing was explained about the synthesis of analogs

 Minor editing of English language required

Reviewer 2 Report

The manuscript entitled “Antiviral effect of 5′-arylchalcogeno-3-aminothymidine derivatives in SARS-CoV-2 infection” discussed the antiviral activity of a series of previously synthesized and published chalcogen thymidine derivatives by the same research group against SARS-CoV-2 (in vitro) and its main protease, Mpro (in silico). The result of the research is interesting; however, some changes are required before publication.

1.      Why did the authors add RdRp to the background statement of the abstract (line 22), although they evaluated the main protease (Mpro) in the in silico study? Why did the authors evaluate Mpro and not RNA-dependent RNA polymerase (RdRp) or even both? Although the MS highlighted the known activity of chalcogens against RdRp? The reason for selecting this viral target should be clarified.

2.      Line 318: Selenium atom would act almost strictly as a linker, displaying favorable size and bond angles. Also, in lines 439-440; Moreover, the role of selenium as an inflammation modulator also opens ways for further research on the benefits of chalcogen-zidovudine in the COVID-19 pathogenesis. ... Did the authors mean that selenium only acts as a linker with no direct role in binding? This means that any linker can substitute Se in the investigated derivatives. Then, the docking appeared useless in explaining the role of selenium in the investigated derivatives. Please references to support this result.

Please see the attached pdf for other comments.

Round 2

Reviewer 1 Report

The content of paper was well organized, all the suggested points are incorporated and easy for the reader to follow the subject discussed, thus support for its acceptance.

Reviewer 3 Report

The authors addressed my previous questions.